# Comparison of Outcomes after Arthroscopic Rotator Cuff Repair between Elderly and Younger Patient Groups: A Systematic Review and Meta-Analysis of Comparative Studies

**DOI:** 10.3390/diagnostics13101770

**Published:** 2023-05-17

**Authors:** Yu-Chieh Hsieh, Liang-Tseng Kuo, Wei-Hsiu Hsu, Yao-Hung Tsai, Kuo-Ti Peng

**Affiliations:** 1Department of Orthopaedic Surgery, Chang Gung Memorial Hospital, Chiayi 61363, Taiwan; exia310208@gmail.com (Y.-C.H.); orma2244@cgmh.org.tw (Y.-H.T.); mr3497@cgmh.org.tw (K.-T.P.); 2Division of Sports Medicine, Department of Orthopaedic Surgery, Chang Gung Memorial Hospital, Chiayi 61363, Taiwan; 7572@cgmh.org.tw; 3School of Medicine, Chang Gung University, Taoyuan 33302, Taiwan

**Keywords:** rotator tear, arthroscopic rotator cuff repair, elderly

## Abstract

This study aimed to compare the outcomes of arthroscopic rotator cuff repair (ARCR) surgery between younger and older patients. We performed this systematic review and meta-analysis of cohort studies comparing outcomes between patients older than 65 to 70 years and a younger group following arthroscopic rotator cuff repair surgery. We searched MEDLINE, Embase, Cochrane Central Register of Controlled Trials (CENTRAL), and other sources for relevant studies up to 13 September 2022, and then assessed the quality of included studies using the Newcastle–Ottawa Scale (NOS). We used random-effects meta-analysis for data synthesis. The primary outcomes were pain and shoulder functions, while secondary outcomes included re-tear rate, shoulder range of motion (ROM), abduction muscle power, quality of life, and complications. Five non-randomized controlled trials, with 671 participants (197 older and 474 younger patients), were included. The quality of the studies was all fairly good, with NOS scores ≥ 7. The results showed no significant differences between the older and younger groups in terms of Constant score improvement, re-tear rate, or other outcomes such as pain level improvement, muscle power, and shoulder ROM. These findings suggest that ARCR surgery in older patients can achieve a non-inferior healing rate and shoulder function compared to younger patients.

## 1. Introduction

Rotator cuff tears (RCTs), one of the most common sources of shoulder pain, can occur due to traumatic injuries, degenerative overuse, or a combination of both [1]. Arthroscopic rotator cuff repair has been popular for some time and provides optimal outcomes after RCT [2]. However, the re-tear rate remains variable, ranging from 7.1 to 46.4%, despite improvements in surgical techniques and implant designs [3,4,5]. In addition, age has been recognized as a significant predictor of increased re-tear rate and poor shoulder function after rotator cuff repair [6]. 

Optimal treatment for rotator cuff tear in the elderly remains the subject of some controversy [7]. Some authors suggest conservative treatment with physical therapy and exercise as the way to achieve pain elimination and functional restoration [8,9,10]. However, surgical treatment has become increasingly popular with the elderly as they seek to improve or restore shoulder function in order to participate in demanding activities [11]. Surgical options for RCTs in the elderly population include reverse total shoulder arthroplasty (RTSA) and arthroscopic rotator cuff repair (ARCR) surgery depending on the size, tendon quality, and reparability of tears. RTSA is indicated for massive irreparable RCT with or without glenohumeral osteoarthritis [12] and provides satisfactory clinical and functional improvements [13,14]. However, it is not very cost-effective [15] and can lead to significant complications [16,17,18]. RTSA may be considered a secondary or salvage procedure after primary ARCR failure, except for those with very poor tendon condition and advanced arthropathy [19].

Concerning ARCR surgery, older individuals usually have several risk factors which may affect outcomes after RCT repair, such as decreased bone density [4], impaired tendon vascularity, progressed fatty infiltration [20], and comorbidities [4,21]. All the above factors may compromise the healing ability of the tendon and hence increase the re-tear risk following RCT repair surgery. Despite the re-tear rate being as high as 25–34% [22], ARCR surgery on rotator cuff tears in older people still produces significant functional improvement and positive changes to the quality of life [23,24,25,26,27], even in patients with massive RCT [25]. Therefore, age may not be the only factor related to poor prognosis in ARCR surgery [26,27]. A recently published systematic review indicated promising clinical outcomes of ARCR surgery in the elderly [28,29]; however, we are unsure how these elderly patients’ outcomes compared with younger patients since the review did not include studies with a younger control [29] and did not have a meta-analysis [28,29].

Moreover, after controlling for sex and tear size, the clinical and structural outcomes of ARCR in elderly patients with symptomatic full-thickness rotator cuff tears are comparable to those in younger patients with at least 1-year follow-up [30]. Thus, our study aimed to summarize the evidence of outcomes after ARCR surgery in elderly patients compared with a younger group and to better understand the clinical outcomes of older patients who received arthroscopic rotator cuff repair. In addition, this study hypothesized that ARCR surgery could achieve non-inferior outcomes in elderly patients compared to younger ones.

## 2. Materials and Methods

### 2.1. Search Strategy and Study Selection

We followed the Preferred Reporting Items for Systematic Reviews and Meta-Analyses (PRISMA) reporting guidelines [31] and the recommendation of the Cochrane collaboration [32] to conduct this systematic review and meta-analysis. We searched online databases including MEDLINE, Embase, Cochrane CENTRAL, and other sources using the keywords: “rotator cuff injury”, “rotator cuff tear”, “seventy”, and “70” for all eligible studies up to 13 September 2022. The details of the search strategy are listed in Appendix A. We also searched the reference list of potentially eligible studies for relevant records. Furthermore, we searched the trial register (www.Clinicaltrial.gov, 13 September 2022) for ongoing trials and consulted experts in this field for ongoing studies. No language limitations were applied. 

### 2.2. Inclusion and Exclusion Criteria

Studies comparing the outcomes of rotator cuff repair surgery in patients older than 65 years with those in a younger group were included in this study. Study designs with a single cohort without a comparative group, review studies, and case series were excluded. Non-human studies were also excluded. 

Two reviewers (YCH and LTK) independently selected the potential studies. First, any irrelevant studies were excluded after checking the titles and abstracts. Second, the full texts of potential studies were checked against the inclusion/exclusion criteria. Third, discrepancies between the reviewers were resolved through discussion, and a third reviewer (WWH) was consulted in cases of disagreement. 

### 2.3. Quality Assessment

We used the Newcastle–Ottawa Scale (NOS) to assess the quality of non-randomized studies [33]. Appraisal criteria included participant selection, comparability, and outcomes. The same two reviewers assessed each of the factors independently, and in case of disagreement, a third reviewer was consulted. 

### 2.4. Data Extraction

Data were extracted from the included studies by two independent authors using the predefined data tables. Relevant data included patient characteristics, sample size, surgical technique, duration of follow-up, and outcomes. The outcomes included functional assessment, re-tear rate, joint range of motion, pain level, and muscle strength. We contacted the relevant author for more information if the data could not be extracted directly from the original studies. If not, we calculated the data and converted them to applicable data. The primary outcome was shoulder pain and function, and the secondary outcomes were re-tear rate, abduction muscle power, shoulder range of motion (ROM), quality of life, and complications. If outcomes were reported at more than one time point, we extracted endpoint data for the following time periods: short-term follow-up (up to six months following surgery), interim follow-up (more than six months and up to one year after surgery), and long-term follow-up (more than one year after surgery). The interim follow-up became the main time point of this review.

### 2.5. Data Analysis

This study used Review Manager 5.4 software (Review Manager Version 5.1.6, Copenhagen: The Nordic Cochrane Centre, The Cochrane Collaboration, 2020) for the meta-analysis. To report the synthetic outcomes, we used an odds ratio (OR) with a 95% confidence interval (CI) and mean difference (MD) with a 95% CI for dichotomous outcomes and continuous outcomes, respectively. A *p*-value of <0.05 was set as the threshold of statistical significance. We used X^2^ and I^2^ statistics to assess statistical heterogeneity with a *p* < 0.10. I^2^ values of 0–24.9%, 25–49.9%, 50–74.9%, and 75–100% were set to indicate no, low, moderate, and high heterogeneity, respectively [32]. We performed subgroup analyses where there was significant heterogeneity or clinical heterogeneity. We also estimated the inter-study variance using tau-squared (τ^2^) statistics [32]. We used a random-effects model meta-analysis due to the inherent clinical heterogeneity of the included studies [34]. We preferred to report improved clinical outcomes when both final outcomes and changes were available. In studies that did not report standard deviations for changes from baseline in continuous variables, we calculated a correlation coefficient from a study with detailed information and used it to impute the standard deviations using the following equation: (γ represents the correlation coefficient [32])
$$SD_{change}=\sqrt{SD_{baseline}{}^{2}+SD_{endpoint}{}^{2}-2\times \gamma \times SD_{baseline}\times SD_{endpoint}}$$

## 3. Results

### 3.1. Included Studies

After searching the above databases for all existing studies, we identified 818 records from MEDLINE, 301 from Embase, and 124 from CENTRAL. One additional record was identified from the reference lists of the eligible studies, and no ongoing trials were identified after searching the trial registers and consulting specialists. A total of 1244 records were identified, from which 58 duplicates were removed. After screening all the records by titles and abstracts, 1161 were excluded. We assessed the remaining 25 full-text articles and excluded 20 records for the stated reasons (Figure 1, Appendix A). Ultimately, five studies were included in the qualitative synthesis and meta-analysis. 

### 3.2. Study Characteristics 

The five studies with a total of 671 participants (197 in the older group and 474 in the younger group) were included in the qualitative systematic review [20,30,35,36,37] (Table 1). The included studies were published between 2010 and 2019, and the enrolled sample sizes ranged from 56 to 238. All the included studies examined the outcomes following rotator cuff repair surgery. An older patient group was compared with a younger group in all of the studies. In addition, there were two prospective cohort studies [20,35], two retrospective cohort studies [36,37], and one retrospective case–control study [29]. The follow-up period of these studies was at least one year. All patients received ARCR, except 30 from Rhee’s study [37], who received open repair. The outcomes and measurements included Constant score, UCLA score, DASH score, re-tear rate, the shoulder joint ROM, shoulder pain severity measured with the visual analog scale (VAS), muscle strength, and quality of life. The details of the included studies are shown in Table 1.

### 3.3. Details of Surgery

The tear size and fatty infiltration of the rotator cuff were documented in the included studies. Four studies [30,35,36,37] reported the tear size using the DeOrio and Cofield classification [38]. Large to massive tears in the included studies ranged from 24% to 58%.

The fatty infiltration was graded by the Goutallier grade [39]. Gwark et al. [30] used the Global Fatty Degeneration Index (GFDI), which is an average Goutallier grade of all four rotator cuff muscles, to report the fatty infiltration status of included patients, while Moraiti et al. [20] reported fatty infiltration of the supraspinatus, infraspinatus, and subscapularis muscles separately. 

About the surgical technique for rotator cuff repair, in the older group, 115 patients received arthroscopic single-row (SR) repair, 76 patients received arthroscopic double-row (DR) or trans-osseous suture equivalent (TOE) repair, and 5 patients received open rotator cuff repair surgery. In the younger group, 262 patients had arthroscopic SR repair, 187 patients had arthroscopic DR or TOE repair, and 25 received open rotator cuff repair surgery (Table 2).

### 3.4. Study Quality

We used the NOS scale to assess the quality of the five included studies. All of the studies achieved three points for selection since the selected controls were all hospital controls. Two studies [20,36] received one point for comparability due to a lack of controls for additional factors other than age. All the studies received three points for assessing exposure except Gwark et al. [30], as their non-response rate differed between the two groups. Details of the studies’ quality assessment can be accessed in Appendix A.

### 3.5. Primary Outcome

All the results are summarised in Table 3. 

#### 3.5.1. Pain Reduction 

Two of the included studies [30,37] with 450 patients reported pain levels measured by VAS. The meta-analysis result showed no significant difference between the two groups concerning the level of pain reduction (MD: −0.33; 95% CI: −0.80 to 0.15; *p* = 0.18; I^2^ = 0%) (Figure 2). Therefore, we are uncertain whether the older patients had better pain reduction than the younger patients after surgery.

#### 3.5.2. Function Improvement One Year after Surgery

Patients ≥ 70 years of age versus patients < 70 years of age.

Three of the included studies [20,30,37] with 530 patients reported shoulder function at one year with Constant scores. The meta-analysis revealed no significant difference in the improvement of the Constant score between the two groups (MD: −0.30; 95% CI: −2.84 to 2.24; *p* = 0.68; I^2^ = 0%, Figure 3). Therefore, we are uncertain whether there was any difference in shoulder function improvement one year after surgery between the two groups. 

Patients ≥ 65 years of age versus patients < 65 years of age.

Osti et al. [36] applied the UCLA score as a function measurement and reported no significant difference between the two groups. Nicholson et al. [35] used the DASH and OSS scores to assess shoulder function. Neither of the two scores showed a significant postoperative difference between the groups. 

### 3.6. Secondary Outcomes

#### 3.6.1. Re-Tear Rate

Patients ≥ 70 years old versus patients < 70 years old.

Three of the included studies [20,30,37] with 530 patients reported a re-tear rate. One used MRI [37], and the other two [20,30] used ultrasound for the evaluation. The result of the meta-analysis revealed there was no significant difference between patients older than 70 and those younger than 70 in the re-tear rate (44/140 vs. 134/390; OR: 0.98; 95% CI: 0.47 to 2.02; *p* = 0.95; I^2^ = 54%) (Figure 4). Therefore, we are uncertain whether there was a difference in the re-tear rate between the two groups. 

Patients ≥ 65 years old versus patients < 65 years old.

Nicholson et al. [35] reported a re-tear rate of 3.6% for patients older than 65 and those younger than 65 (1/28 vs. 2/56). There was no statistically significant difference in the re-tear rate between the two groups. 

#### 3.6.2. Abduction Muscle Power

Patients ≥ 70 years old versus patients < 70 years old.

Two studies [30,37] reported abduction muscle power, measured by a handheld digital myometer. Meta-analysis revealed no significant difference between those older and younger than 70 years old (MD: 0.31 Kg; 95% CI: −0.13 to 0.75 Kg; *p* = 0.17; I^2^ = 0%) (Figure 5).

Patients ≥ 65 years old versus patients < 65 years old.

None of the included studies reported this outcome. 

#### 3.6.3. Shoulder Range of Motion Improvement

Patients ≥ 70 years old versus patients < 70 years old.

Three of the included studies [20,30,37] with 530 patients reported shoulder ROM as one of the outcomes. The result showed there was no difference in the improvement of shoulder ROM between the groups older and younger than 70 in terms of forward flexion (MD: −3.29; 95% CI: −9.00 to 2.24; *p* = 0.26; I^2^ = 0%) and abduction (MD: 4.05; 95% CI: −2.35 to 10.45; *p* = 0.21; I^2^ = 0%). The group of patients older than 70 had a better improvement in terms of external rotation than the group younger than 70 (MD: −1.15; 95% CI: −2.11 to −0.19; *p* = 0.02; I^2^ = 0%) (Figure 6).

Patients ≥ 65 years old versus patients < 65 years old.

Osti et al. [36] reported no significant difference between groups of over and under 65 in forward flexion, external rotation, and internal rotation. 

#### 3.6.4. Quality of Life

Osti et al. [36] assessed the quality of life using the SF-36 self-administered questionnaire. It showed a significant improvement compared to the preoperative scores, but no significant difference in all categories between the groups over and under 65 years old. Nicholson et al. [35] demonstrated an EQ-5D score for assessing the quality of life. There was no significant difference after 12 months and 24 months postoperatively between the groups over and under 65 years old.

#### 3.6.5. Complications

None of the included studies reported this outcome.

## 4. Discussion

This study’s principal findings indicated no significant difference between the older and younger patient groups in terms of pain and shoulder function improvement. These two groups had similar re-tear rates, abduction muscle power, quality of life, and complications within two years after ARCR. They also showed no significant difference in shoulder ROM, except in the degree of external rotation, which failed to reach clinical significance. In sum, ARCR in senior patients had a non-inferior outcome compared to younger patients.

This study further indicated that the improvement in the Constant score was equivalent between the two groups. This finding was consistent with the previous studies [5,40]. Witney-Lagen et al. [40] reported that elderly patients benefited as much from ARCR as the younger patients. Similar improvements in shoulder function, pain, and satisfaction were also observed in both elderly and control patients. A recently published database study [5] reported that the improvement in shoulder function from baseline was no different between patients aged ≥70 vs. <70 years, although patients aged ≥70 years had a significantly lower shoulder function compared with those <70 years. Moreover, Kukkonen et al. [41] demonstrated an estimated minimum clinically important difference of 10.4 points as the threshold for the Constant score in patients with rotator cuff tears. All the enrolled studies had a mean improvement of Constant score over this threshold, which indicated that all the patients had clinically significant improvements in shoulder function after the surgery. 

The synthetic results in the current study showed no significant difference in re-tear rate between those older than 70 years and those younger than 70 years. Nicholson et al. [35] reported a similar result: the re-tear rate was not different between those older than 65 years and those younger than 65 years. However, one of the included studies [20] reported a tendency for older individuals to have a higher re-tear rate, contrary to the other included studies. As the rehabilitation protocols varied across the five studies, the period of postoperative immobilization may have influenced the surgical outcome. Moraiti et al. started an actively-assisted range of motion program at four weeks postoperatively [20], while the other two studies started six weeks after surgery [30,37]. The period of immobilization after rotator cuff repair remained under debate [42,43,44,45]. Hsu et al. [42] indicated no difference in the healing rate between early and delayed rehabilitation after rotator cuff repair. Keener et al. [43] enrolled 124 patients to compare the outcome between two groups receiving either early ROM or delayed ROM within six weeks after index surgery, whereby they found no difference in the healing rate. Notably, Keener only enrolled patients under the age of 65 with small and median tear sizes. Koh et al. [44] performed a randomized controlled trial comparing the effect of 4 weeks and eight weeks of immobilization, which yielded no evidence that a poor healing rate was correlated with a shorter period of immobilization. Additionally, a systematic review published in 2017 indicated that early ROM treatment improves ROM but increases the risk of rotator cuff re-tear and suggested the rehabilitation protocol should be modified according to the tear size [45]. Notably, none of these studies enrolled an aged population [42,43,44,45], which made the judgment of the impact of rehabilitation on the functions and outcomes of the elderly difficult. Since the poor healing potential of aged tendons remains the major concern in this age group, further studies on the elderly may be warranted to address this issue. 

Various cuff repair approaches were used in the included studies, including arthroscopic and open repair. Debates persist regarding the choice between arthroscopic and open rotator cuff repair [46,47,48]. Some previous studies have shown no difference between open and arthroscopic repair outcomes. Severud et al. [46] report a long-term retrospective comparison between ARCR and mini-open repair, indicating that ARCR produces outcomes and complications comparable to mini-open repair. Bayle et al. [47] found no advantages of arthroscopic surgery over open cuff repair surgery regarding the clinical outcomes and cuff integrity at 1-year follow-up. A recent meta-analysis [48], comprising six randomized controlled trials, revealed that ARCR and mini-open cuff repair surgery showed no difference in function, pain, and ROM at 3-, 6- and 12-month follow-ups. Thus, the difference in approaches among the studies included in the current study may be ignored.

Different types of repair techniques were also noted in the included studies. However, we opted not to perform subgroup analysis due to the lack of detailed data in each study. Xu et al. [49] included nine studies in their meta-analysis to compare the outcomes of SR and DR repair techniques. They report that the DR technique is superior to SR in terms of a lower re-tear rate, higher American Shoulder and Elbow Surgeons (ASES) score, and greater ROM, especially in tears larger than 30 mm. Another meta-analysis from Sobhy et al. [50] concluded that DR had a better UCLA score and cuff structural integrity. Hein et al. [51] performed a systematic review to compare the re-tear rate of SR, DR, and TOE techniques. Both DR and TOE had a lower re-tear rate than SR. Plachel et al. [52] demonstrated a long-term outcome follow-up, comparing SR and DR. The study reports no difference in terms of Constant score, Western Ontario Rotator Cuff index, and Subjective Shoulder Value between SR and DR. Despite SR having a higher re-tear rate than DR, clinical outcomes were not affected. Ying et al. [53] report no significant difference between the SR and DR groups in terms of Constant or ASES scores; however, DR was better in UCLA scores, tendon healing, and ROM. Overall, the study remains inconclusive about the superiority of particular techniques. Thus, the impact of the difference in cuff repair techniques across the included studies may be ignored unless updated evidence changes the current conclusion on this issue.

This study had several limitations. First, various cuff repair techniques were performed in the included studies, and individual data about surgical techniques were not provided. Furthermore, additional procedures, such as acromioplasty, and tenodesis/tenotomy of the biceps tendon, were not evenly distributed across the included studies. As a result, it is difficult to assess the impact of these different procedures on the outcomes in the two groups, although the differences in surgical methods have rarely been shown to significantly influences clinical outcomes [54,55]. Third, the included studies did not discuss repair conditions. Since the degree of repair integrity is a predictor of shoulder functions after arthroscopic rotator cuff repair [56], future studies on this issue should also document the status of cuff repair. Fourth, since the included studies had follow-up no longer than two years, this study’s evidence showing the older group had a similar re-tear risk to the younger group can only apply within two years after cuff repair surgery. Future studies with a longer follow-up (>2 years) could address the mid-term outcomes in patients over 70 after arthroscopic repair surgery. Finally, none of the included studies reported the degree of cuff tear arthropathy before surgery. Since the degree of concomitant arthropathy is a significant predictor of repair failure and subsequent arthroplasty in this age group [57], we suggest the severity of cuff arthropathy should be recorded and followed. Future studies are needed, but in order to optimize the research effects, they should have identical surgical techniques, recording repair status, and the severity of cuff arthropathy with a longer follow-up period (≥2 years) to minimize the uncertainty from the heterogeneities of the currently available evidence.

## 5. Conclusions

We compared the outcomes between older and younger patients after receiving ARCR surgery. ARCR surgery among older patients had clinical outcomes and functions comparable to those in younger patients. In selected cases, ARCR surgery could achieve non-inferior results in senior patients, compared to younger ones.

## Figures and Tables

**Figure 1 diagnostics-13-01770-f001:**
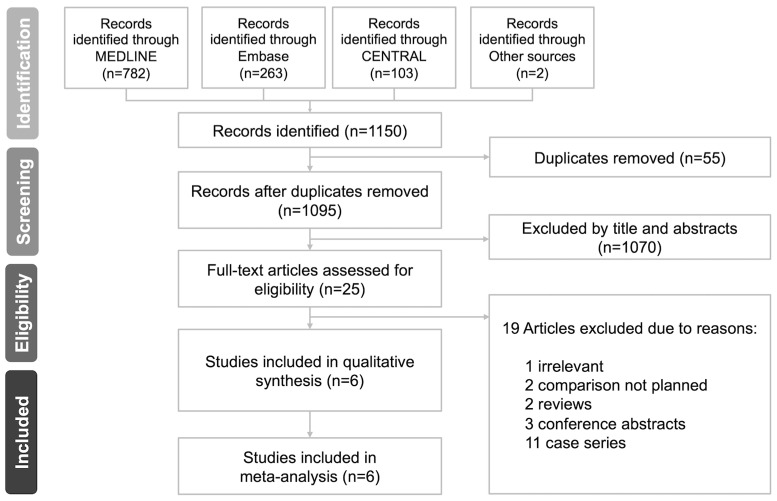
PRISMA flow diagram of the study.

**Figure 2 diagnostics-13-01770-f002:**
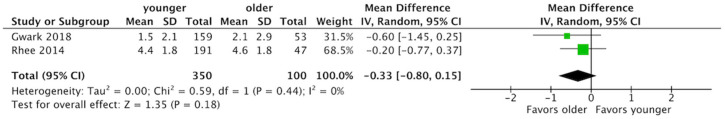
Forest plot of pain reduction [30,37]. There was no significant difference between the two groups concerning the level of pain reduction (MD: −0.33; 95% CI: −0.80 to 0.15; *p* = 0.18; I^2^ = 0%). MD, median difference; CI, confidence interval.

**Figure 3 diagnostics-13-01770-f003:**
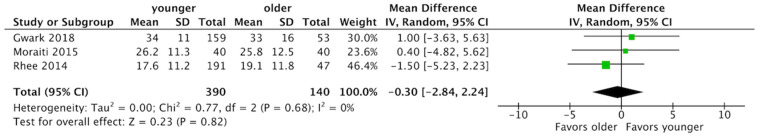
Forest plot of shoulder function improvement by Constant score [20,30,37]. There was no significant difference between the two groups concerning improvement of Constant score (MD: −0.30; 95% CI: −2.84 to 2.24; *p* = 0.68; I^2^ = 0%). MD, median difference; CI, confidence interval.

**Figure 4 diagnostics-13-01770-f004:**
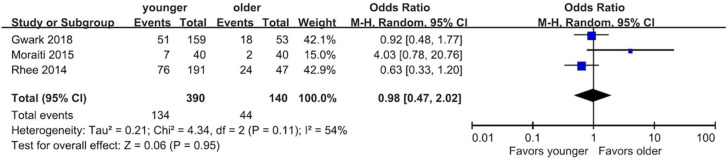
Forest plot of the re-tear rate [20,30,37]. There was no significant difference in re-tear rate between the two groups (OR: 0.98; 95% CI: 0.47 to 2.02; *p* = 0.95; I^2^ = 54%). CI, confidence interval; OR, odds ratio.

**Figure 5 diagnostics-13-01770-f005:**
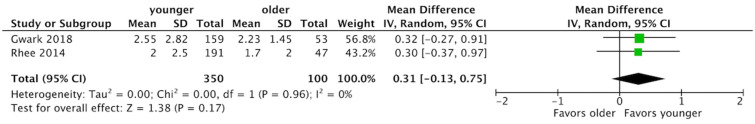
Forest plot of the abduction muscle power [30,37]. There was no significant difference in the abduction muscle power between the two groups (MD: 0.31 Kg; 95% CI: −0.21 to 0.98 Kg; *p* = 0.17; I^2^ = 0%). CI, confidence interval; MD, mean difference.

**Figure 6 diagnostics-13-01770-f006:**
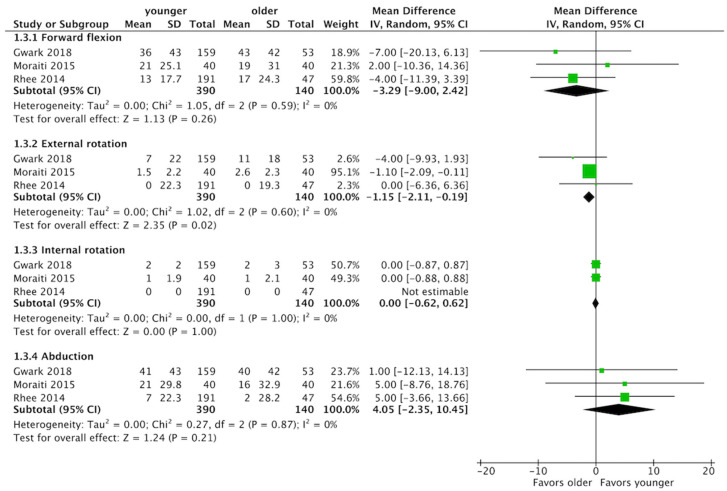
Forest plot of the shoulder ROM improvement [20,30,37]. There was no significant difference in ROMs between the two groups confidence interval; ROM, range of motion.

**Table 1 diagnostics-13-01770-t001:** Characteristics of included studies.

Study (Author, Year)	Country	Study Design, LOE	Inclusion/Exclusion Criteria	Sample Size (M/F)	Outcomes	NOS	Funding
Osti, 2010 [36]	Italy	Retrospective cohort, II	Inclusion: FTRCT with cuff repair surgery Exclusion: Advanced rotator cuff arthropathy (Hamada Grade 3–5), severe fatty infiltration (Goutallier stage 3 or 4), subscapularis tear, fracture, instability, previous surgery, etc.	Age > 65: 28 (19/9) Age < 65: 28 (19/9)	Function (UCLA score), shoulder ROM, QoL (SF-36)	7	NR
Rhee, 2014 [37]	Korea	Retrospective cohort, II	Inclusion: FTRCT with ARCR surgery Exclusion: AC joint OA requiring resection, advanced GH OA, biceps lesion, revision surgery	Age > 70: 47 (14/33) Age < 70: 191 (74/117)	Function (Constant score, UCLA score), re-tear rate, shoulder ROM, pain level (VAS), muscle strength	7	NR
Moraiti, 2015 [20]	France/ Greece	Prospective cohort, II	Inclusion: FTRCT with ARCR surgery Exclusion: Previous trauma or surgery, advanced GH joint OA	Age > 70: 41 Age < 70: 40	Function (Constant score), re-tear rate, shoulder ROM	7	NR
Gwark, 2018 [30]	Korea	Retrospective case–control, III	Inclusion: FTRCT with ARCR surgery Exclusion: previous trauma history	Age > 70: 53 (19/34) Age < 70: 159 (57/102)	Function (Constant score), re-tear rate, shoulder ROM, pain level (VAS), muscle strength	7	NR
Nicolson, 2019 [35]	United Kingdom	Prospective case–control, III	Inclusion: FTRCT with ARCR surgery Exclusion: Previous Injury, revision surgery, partial tear, unable to complete repair	Age > 65: 28 (17/11) Age < 65: 56 (28/28)	Function (DASH score, OSS score), QoL (EQ-5D), re-tear rate	7	Yes

AC: acromioclavicular; ARCR: arthroscopic rotator cuff repair; DASH: disabilities of the arm shoulder and hand questionnaire; EQ-5D: EuroQol five-dimension questionnaire; FTRCT: full-thickness rotator cuff tear; GH: glenohumeral LOE: level of evidence; NOS: Newcastle–Ottawa Score; NR: not reported; OA: osteoarthritis; OSS: Oxford shoulder score; QoL: quality of life; ROM: range of motion; VAS: visual analog.

**Table 2 diagnostics-13-01770-t002:** Details of surgery.

Study	Group	Tear size ^a^ (I/II/III or Small Medium/Large/Massive)	Fatty Infiltration ^b^	Repair Method (SR/DR or TOE)
Osti, 2010 [36]	Age > 65	3/15/10	SSP (18/10) ^c^ ISP (23/5) SSC (24/4)	All single row
	Age < 65	5/16/7	SSP (19/9) ISP (25/3) SSC (25/3)	
Rhee, 2014 [37]	Age > 70	22/9/16	NR	27/15
	Age < 70	85/68/38		101/65
Moraiti, 2015 [20]	Age > 70	NR	SSP 1.9 ± 0.1 ISP 1.6 ± 0.2 SSC 1.6 ± 0.2	22/18
	Age < 70		SSP 0.8 ± 0.1 ISP 1.0 ± 0.3 SSC 0.75 ± 0.3	15/25
Gwark, 2018 [30]	Age > 70	22/15/16	GFDI ^d^ 1.6 ± 0.3 ISP 2.2 ± 0.7	38/15
	Age < 70	66/45/48	GFDI 1.7 ± 0.5 ISP 2.0 ± 0.8	118/41
Nicholson, 2019 [35]	Age > 65	9/19 large or massive	NR	All double row
	Age < 65	23/33 large or massive		

DR: double row; GFDI: global fatty degeneration index; ISP: infraspinatus; NR: not reported; SR: single row; SSP: supraspinatus; SSC: subscapularis; TOE: transosseous equivalent. ^a^ evaluated by Boileau classification system (MRI) or DeOrio and Cofield classification (intraoperative). ^b^ evaluation by Goutallier classification. ^c^ expressed as Grade 0–1/Grade 2. ^d^ Global fatty degeneration index, GFDI: the average Goutallier grade of the supraspinatus, infraspinatus, subscapularis, and teres minor muscles.

**Table 3 diagnostics-13-01770-t003:** Summary of synthetic results.

Outcomes	No. of Studies, n	Patients in Younger Group, n	Patients in Older Group, n	OR/MD (95% CI)	*p*	I^2^
Pain level (VAS) improvement	2	350	100	−0.33 (−0.80, 0.15)	0.18	0%
Constant score improvement	3	390	140	−0.30 (−2.84, 2.24)	0.82	0%
Re-tear rate	3	390	140	0.98 (0.47, 2.02)	0.95	54%
Muscle power	2	350	100	0.31 (−0.13, 0.75)	0.17	0%
Joint ROM improvement						
Forward flexion	3	390	140	−3.29 (−9.00, 2.42)	0.26	0%
External rotation	3	390	140	−1.15 (−2.11, −0.19)	0.02	0%
Abduction	3	390	140	4.05 (−2.35, 10.45)	0.21	0%

CI: confidence interval; MD: mean difference; OR: odds ratio; ROM: range of motion; VAS: visual analog scale.

## Data Availability

All data were publicly available. Data sharing is not applicable to this article.

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
