# Peer review of "Comparison of Outcomes after Arthroscopic Rotator Cuff Repair between Elderly and Younger Patient Groups: A Systematic Review and Meta-Analysis of Comparative Studies"

_diagnostics, 2023, doi:10.3390/diagnostics13101770_

Round 1

Reviewer 1 Report

  The purpose of this study is to evaluate the efficacy of arthroscopic rotator cuff repair in elderly patients by means of a meta-analysis of articles in the literature.
This is the first meta-analysis evaluating the results of studies of arthroscopic cuff repair with a group of young patients and a group of patients older than 65-70 years. The methods are well described and adequate The conclusions are consistent with the results of the meta-analysis and answer the main question posed by the study The references appropriate, but it is necessary to correct the references in table 1 and 2. line 32 remove T Correct the references in Table 1 and 2 referring to the studies selected for the meta-analysis. line 287-289: remove: This section may be divided by subheadings. It should provide a concise and precise description of the experimental results, their interpretation, as well as the experimental conclusions that can be drawn.Perhaps it refers to some of your notes. Line 370-371: 'However, the difference ... clinical influence'. This statement is not understandable, please explain it better.

Minor editing of English language required

Author Response

Please see the attachment."

Reviewer 2 Report

Review

lines 13 and 14 The aim was to compare the outocomes of rotator cuff repair between young an elderly patients.

Lines 17 and 18 merge

Line 32 delete T, and tears in plural

Line 33 RC tears in younger patients are also degenerative or due to chronic overuse and do not originate only from a single trauma episode

Line 36 please provide the retear rate incidence refering to more than 1 papers.

Line 45 RSA is not an indication for the treatment of RC tears but only in irreparable tears or arthritis. Please rephrase

Line 53 complicated comorbidities. Please correct

Line 55 provide numeric data

line 56 on rotator cuff injuries. Not correct, please improve

Line 74 Cochrane collaboration. Citation is necessary

Line 84 define the age limit. Is it 65 or 70 years?

Minor typing and spelling errors should be corrected as described in the previous section.
